Age-related variation in the anthropometric profiles, body composition and functional capacities of female soccer players

Toselli Stefania
Moro Federica federica.moro10@unibo.it
Perugini Martina
http://orcid.org/0000-0002-8171-5096 Mauro Mario
Life Quality Studies, University of Bologna , Bologna , Italy
Plavec Davor
Electronic publication date: 2025 Oct 30
Publication date: 2025
Volume: 13
Electronic Location ID: e20096
Received 2025 Jul 18; Accepted 2025 Aug 27
Copyright: © 2025 Toselli et al.
Copyright year: 2025
Copyright holder: Toselli et al.
License: This is an open access article distributed under the terms of the Creative Commons Attribution License, which permits unrestricted use, distribution, reproduction and adaptation in any medium and for any purpose provided that it is properly attributed. For attribution, the original author(s), title, publication source (PeerJ) and either DOI or URL of the article must be cited.
License URL: https://creativecommons.org/licenses/by/4.0/

Keywords: Physiology, Performance, Maturation, Somatic

Funding: The authors received no funding for this work.

==============================
This study aimed to identify and compare the basic anthropometric characteristics, physical performance, and game-related skills of female soccer players across different age groups, including under-15 (U15), U17, U19, and first teams. Also, it aimed to account for performance tests variability through anthropometric features. A total of 56 athletes participated in a comprehensive test battery assessing anthropometric and bioelectrical parameters, air displacement plethysmography (ADP), and physical performance (Countermovement Jump Test (CMJ) and 30-15IFT). The older groups outperformed the younger players in physical and physiological characteristics. Significant differences were observed in body dimensions and composition, with the first team showing higher fat-free mass (FFM) than U15 and U17 players. Skinfold thickness was consistent across groups. The first team also exhibited the highest levels of power and speed, while U15 athletes showed the lowest CMJ. General linear models by stepwise procedures identified training hours as the main predictor for CMJ and speed, with each additional hour of training improving CMJ by 1 cm and speed by 1 km/h. Additionally, a decrease in skinfold thickness predicted improved power. A multidimensional approach, including anthropometric, bioelectrical, and physical performance data, provides essential insights for supporting athletic development in young female soccer players and can inform tailored training strategies to enhance their physical performance.

Introduction

Over the last decades, female soccer has seen a significant increase in participation and increased financial support from governing bodies. However, information regarding the physical characteristics and the health and performance needs of female soccer are still scarce (Randell et al., 2021).

One of the most important challenges in sports is the identification and development of young athletes with high potential, given the intense competition in high-performance sports. However, data on female talent identification is minimal compared to male ones. This challenge is also influenced by the inaccuracy of current talent identification systems despite substantial investments in their development (Koz, Fraser-Thomas & Baker, 2012). In this context, modern approaches consider the multidimensional nature of sports talent, simultaneously addressing the needs of team sports and aiming to increase the accuracy and success of talent identification processes (Vaeyens et al., 2008). Female soccer players required to have a combination of physical and psychological characteristics to be competitive at the highest level. These characteristics are essential not only for selection processes but must also be considered for injury prevention and continuous performance improvement. Monitoring anthropometry and body composition, training data and/or physical tests changes is key for the structured development of performance and the prevention of overuse injuries (Lesinski et al., 2017; Oliveira et al., 2021). These data are essential to help coaches to evaluate their training on a daily basis by tailoring ongoing decision-making processes (Bourdon et al., 2017). In female soccer, as in other sports, body composition assessment provides health and performance relevant information and allows for planning possible exercise and dietary interventions (Randell et al., 2021). In the assessment of body composition, special attention is typically given to body fat mass (FM) and fat-free mass (FFM), since an increase in fat mass impairs performance, while an increase in muscle mass can promote the development of strength and power, which are important for player performance (Collins et al., 2021; Silvestre et al., 2006). Data on female soccer players are scarce; literature affirms that body fat percentage ranges between 18.0% and 28.2% in players aged 10 to 16 years and from 14.5% to 22% for elite adult players (Malina et al., 2021; Randell et al., 2021). Recently, attention has also been paid to other body composition variables, such as total body water (TBW), intracellular water (ICW), and extracellular water (ECW), to monitor the hydration status of athletes. ICW, in particular, is a good predictor of strength and power in athletes (Silva et al., 2011, 2014). In addition, fluid content detected by bio-impedance analysis (BIA) appeared able to predict the incidence of lower-limb muscles injury (Chen et al., 2024). Considering that recent approaches such as vectorial BIA or bioelectrical impedance vector analysis (BIVA) simply provide indications of the ICW/ECW ratio (Gonzalez et al., 2016; Norman et al., 2012), speculating on vector length and displacement due to individuals and time may favor interesting insight of water compartmentation (Norman et al., 2012; Piccoli et al., 1994).

However, whilst body performance and composition are divided for experimental purposes, elite players need specific body characteristics to reach best field goals. To date, soccer is characterized by many physiological requirements such as speed and agility, strength and power, and endurance in both male and female tournaments (Buchheit, 2021). Strength and power, particularly that of the lower limbs, are essential to generate speed, acceleration, and power in shots and jumps (Markovic & Mikulic, 2010), while agility, which includes the ability to change direction and speed quickly, is a determining factor in soccer performance (Girard, Mendez-Villanueva & Bishop, 2011). In addition, due to match duration is around 90 min, endurance and fatigue tolerance are essential to prolong high degree of performance by reducing injury risks (Buchheit, 2021).

Understanding the key characteristics for successful female soccer performance is crucial for coaches, physicians, nutritionists, and exercise physiologists to better support player selection and training programs. Given the need to explore the relationship between anthropometry, body composition, and training demands in young female players for optimal development and performance, the aims of the present study were:

(1) to describe and compare the basic anthropometric characteristics, body composition and physical performance outcomes of different ages female soccer players selected from an elite Italian team;

(2) to discover and report linear correlations between anthropometrical profiles and performance achieved;

(3) to draw generalized linear regression models that predict physical performance variability from body characteristics.

Materials and Methods

Portions of this section were previously published as part of a preprint (Mauro, Moro & Toselli, 2025) and doctoral thesis (Mauro, 2025).

Study design and participants

A cross-sectional study was carried out on a sample of 56 female soccer athletes (from the Under 15 to first team) belonging to the professional Italian soccer team Bologna Football Club 1909 participating in the first division. Before the evaluations, each participant or parent was informed of all procedures and risks of the study protocol. All participants or parents provided a written informed consent before the study began. Each participant was asked to fast for at least 4 h, and to avoid any training at least 12 h before the test. The anthropometry and body composition evaluations were conducted at the Sports Laboratory of the University of Bologna, from 11 a.m. to 5 p.m., under conditions of approximately 19.8 °C and 51% humidity, while the performance tests were carried out at the Football Club sports center, on a different day, on a football field with a grassy surface to simulate game conditions. The study was approved by the Bioethics Committee of the University of Bologna (Approval code: 25027).

Assessment of age at menarche and anthropometric measurements

A face-to-face interview was conducted by the same operator to collect information on participants. The girls were asked if menarche had been reached. Respondents who provided affirmative answers were then requested to recall the precise date of their first menstruation (day, month, and year). Anthropometric measurements, including stature, weight, lengths, widths, circumferences, and skinfold thicknesses, were collected by a trained operator following the standardized procedures (Norgan, 1988). Stature and sitting height were measured to the nearest 0.1 cm using a GPM stadiometer (Zurich, Switzerland), with leg length calculated by subtracting sitting height from height. The body mass was measured to the nearest 0.1 kg using a calibrated electronic scale, with participants wearing light indoor clothing and being barefoot. Circumferences for the relaxed and contracted upper arm, waist, hip, thigh, and calf were taken using a non-stretchable tape measure to the nearest 0.1 cm (Norgan, 1988). Bone widths at the humerus and femur were measured to the nearest 0.1 cm with a sliding caliper. Skinfold thicknesses at nine sites (biceps, triceps, subscapular, supraspinal, sovrailiac, abdominal, thigh, and lateral and medial calf) were assessed using a Lange skinfold caliper (Beta Technology Inc., Houston, TX, USA) with precision to the nearest 1 mm (Norgan, 1988). Santos et al. (2014) sport-specific reference percentiles were used to evaluate the sum of skinfold thicknesses (triceps, subscapular, biceps, sovrailiac, abdominal, thigh, and medial calf), appendicular skinfolds, and trunk skinfolds. Body mass index (BMI) was calculated using the standard formula: weight (kg)/stature squared (m2). Cross-sectional areas, including total area (total upper area (TUA); total calf area (TCA)), muscle area (upper muscle area (UMA); calf muscle area (CMA); total upper area (TUA)), and fat area (upper fat area (UFA); calf fat area (CFA); total fat area (TFA)) for the upper arm, calf, and thigh, were calculated according to Frisancho (1990). Additionally, the arm fat index (AFI) and calf fat index (FCI) were calculated.

Bioelectric impedance analysis (BIA)

The impedance measurements were performed using bioimpedance analysis (BIA 101 Anniversary; Akern, Florence, Italy) with an electric current at 50 kHz. Measurements were made with four electrical conductors; the subjects were positioned supine with a leg opening of 45° relative to the median line of the body and the upper limbs placed 30° away from the trunk. After the skin was cleansed with alcohol, two Ag/AgCl low-impedance electrodes (Biatrodes Akern Srl, Florence, Italy) were placed on the back of the right hand, and two electrodes were placed on the corresponding foot (Piccoli et al., 1994). To avoid disturbances in fluid distribution, athletes were instructed to abstain from food and liquids for ≥4 h before the test (Kyle et al., 2004). A normal breakfast was consumed by the athletes at 07:00, and the first measurement was taken at 11:00. Total body water (TBW), extracellular water (ECW), and intracellular water (ICW) were estimated based on the method of Matias et al. (2016). Vector length (VL) was calculated as (adjusted R2 + adjusted Xc2) 0.5, and phase angle (PA) was calculated as the arctangent of Xc/R × 180°/π. BIVA was carried out using the classic methods, e.g., normalizing R (Ω) and Xc (Ω) for height in meters (Piccoli et al., 1994).

Air displacement plethysmography (ADP)

The ADP BOD POD GS-X (Cosmed, Rome, Italy) measured the participant’s body volume (BV) throughout the Boyle’s law gas principles (Dempster & Aitkens, 1995). The manufacturer’s recommendations were applied to calibrate the instrument before each assessment. Firstly, the environmental temperature was lower than 27 °C, while the humidity was about 40%. An autorun and a BV trial tested the air volume contained in to both chambers. The BOD POD GS-X body mass scale was calibrated at the beginning of the evaluation day. Minimal clothes (a tight-fitting swimsuit and cap) were worn by all participants, and jewelry or glasses were not worn for the test. Each participant was instructed to adequately sit within the chamber and was asked to avoid any unnecessary body movement during the trial. One test encompassed two trials, while a further trial was performed whether the previous varied by more than 1%. The ADP Omnia software used the measured body mass and volume to calculate the body density and then convert it into body fat (%) using the Brozek equation (Brožek et al., 1963). Fat mass (FM) and fat-free mass (FFM) were derived.

Physical performance

Measures included the Countermovement Jump Test (CMJ) and 30-15IFT.

Before each trial, a standardized warm-up of 10 min with general and specific exercises was performed by the participants (Buchheit, 2008). The CMJ was first assessed for its greater contribution to the neuromuscular system, while the 30-15IFT followed.

The CMJs were measured using two photocells (OptoJump®, Microgate, 11 Miller Road, 10541 Mahopac (NY)—U.S.A.). To test the CMJ, each participant was instructed to perform vertical jumps (CMJ) from an erect standing position with a knee angle of 180°. A countermovement down to approximately a 90° knee angle was performed (Ingebrigtsen et al., 2014). The highest values from three attempts were used for analysis. As regards the 30-15IFT, all procedures were assessed in accordance with Buchheit (2008). The subjects were required to run between two lines set 40 m apart, with an acoustic signal used to help them reach and maintain the desired speed. The test consisted of 30-s shuttle runs interspersed with 15-s passive recovery intervals. Baseline speed was set at 8 km·h−1 and was increased by 0.5 km·h−1 every 45-s stage thereafter. The targeted distances to run were calculated based on an empirical value of 0.7 s from the 30-s running periods for each change of direction. During recovery, the subjects walked forward to the closest line from where the next run stage was started. The subjects were instructed to run until they could no longer maintain the required speed. The last speed recorded was used as the maximal running speed (MRS30-15IFT). The MRS30-15IFT exhibited large correlation degrees with maximal oxygen consumption, heart rate and blood lactate variation, and maximal speed (Buchheit, 2008; Paravlic et al., 2022).

Statistical analysis

Descriptive statistics were described as mean ± standard deviation. The Pearson correlation coefficient (r) was computed to evaluate linear correlation between continuous variables, with a type I error probability of 5% associated to the null hypothesis of no relationship (r = 0). The normality of distribution was verified by the Kolmogorov-Smirnov test, whereas the assumption of equality of variance was verified using the Levene test. ANOVA was used to examine differences between the different groups. Furthermore, Tukey adjustments were made for post-hoc comparisons. The effect size was evaluated using partial eta squared (η2) and classified as: no effect = 0 to 0.039, minimum effect = 0.04 to 0.24, moderate effect = 0.25 to 0.63, and strong effect = ≥0.6433 (Cohen, 2013).

Backward multiple regression was used to estimate the relative contributions of anthropometric, body composition, and impedance parameters to the variability of individual physical performance tests. The significance level for removing regressors from the model was set at 5%, and both Akaike information criteria (AIC) and adjusted goodness of fit (adj.-R2) were used for determining the best model.

The significance of the effects was set at p < 0.05 for all the analyses. All calculations were performed using STATISTICA software (v.8, StatSoft, Tulsa, OK, USA).

Results

Table 1 shows soccer experience, the onset of menarche, anthropometric characteristics, and body composition in female soccer athletes of the present study. The years of experience differed between U15 athletes and the first team, while training hours differed between the latter and all other groups: younger participants trained for about 6 h per week, with a two-increasing hours per category up to 12 (first team).

Table 1 Soccer experience: onset of menarche, anthropometric characteristics, body composition, in female soccer athletes.

	U15 (12)	U17 (9)	U19 (10)	First team (25)				
	Mean	SD	Mean	SD	Mean	SD	Mean	SD	F	p	η2	
Age (year)	14.03	0.58	15.90	0.89	17.36	0.76	24.56	3.59	5.55	0.00	0.25	
Begin age (year)	9.08	2.23	9.00	2.78	8.60	2.88	8.08	3.33	0.41	0.75	0.02	
Exp (year)	4.95	2.26	6.90	2.74	8.76	2.54	16.48	5.28	28.61	–	0.62	
Training hour	6.17	0.58	8.00	–	8.00	–	12.00	–	1567.99	–	0.99	
Menarche age (year)	11.70	1.57	12.56	1.01	12.90	1.52	13.64	1.22	5.55	0.00	0.25	
BM (kg)	53.80	7.03	57.51	5.66	58.23	6.08	62.21	6.89	4.64	0.01	0.21	
Stature (cm)	159.79	4.97	164.53	4.89	164.23	4.95	166.76	6.32	4.16	0.01	0.19	
Trunk h (cm)	83.48	2.45	87.31	3.59	87.47	2.05	88.14	3.00	7.52	0.00	0.30	
Low limb lenght (cm)	76.31	3.02	77.22	2.76	76.76	3.25	78.61	4.10	1.42	0.25	0.08	
Arm rel (cm)	24.75	2.59	26.32	2.91	25.83	1.38	26.97	1.75	3.12	0.03	0.15	
Arm stret (cm)	25.78	2.48	27.27	2.56	26.73	1.36	28.42	1.77	5.08	0.00	0.23	
Hip (cm)	67.39	3.43	67.37	2.83	68.21	3.26	72.22	3.71	8.24	0.00	0.32	
Thigh (cm)	49.83	3.39	51.23	2.80	53.41	4.42	53.90	3.32	4.34	0.01	0.20	
Calf (cm)	34.07	2.17	34.93	2.78	36.35	2.82	35.88	1.84	2.48	0.07	0.13	
Humeral (cm)	5.92	0.29	5.89	0.31	5.82	0.27	5.90	0.28	0.25	0.86	0.01	
Femoral (cm)	9.11	0.51	8.90	0.43	9.68	0.73	9.42	0.70	3.03	0.04	0.15	
Biceps SK (mm)	7.25	2.01	7.19	2.90	6.55	1.52	6.33	2.66	0.53	0.66	0.03	
Triceps SK (mm)	12.71	2.32	12.57	2.63	12.40	3.10	11.93	2.88	0.27	0.85	0.02	
Subscapular SK (mm)	8.06	2.00	8.96	1.53	9.63	1.45	8.85	2.03	1.32	0.28	0.07	
Suprailiac SK (mm)	12.06	3.05	11.63	2.46	13.80	2.97	10.70	3.49	2.34	0.08	0.12	
Supraspinal SK (mm)	8.03	2.78	7.93	2.32	8.80	3.21	6.79	2.36	1.66	0.19	0.09	
Abdominal SK (mm)	9.39	2.40	8.70	1.65	9.82	3.22	9.19	2.24	0.36	0.78	0.02	
Thigh SK (mm)	15.32	3.17	16.19	2.30	15.73	3.19	16.79	3.25	0.70	0.56	0.04	
Medial calf SK (mm)	10.86	2.59	10.19	2.53	11.03	3.23	9.07	2.83	1.74	0.17	0.09	
Lateral calf SK (mm)	11.33	2.60	11.50	3.03	13.20	4.10	10.83	2.85	1.42	0.25	0.08	
R (ohm)	568.35	58.63	578.03	46.97	558.33	71.59	508.05	36.28	6.70	0.00	0.28	
Xc (ohm)	58.78	7.28	65.91	5.64	64.63	6.73	63.72	6.43	2.54	0.07	0.13	
PhA	5.83	0.64	6.41	0.26	6.55	0.40	7.09	0.66	13.94	0.00	0.45	
D (kg/L)	1.04	0.01	1.05	0.01	1.05	0.01	1.05	0.01	1.90	0.14	0.10	
BF (%)	23.53	3.10	21.77	4.59	21.71	4.18	20.12	4.33	1.92	0.14	0.10	
FM (kg)	12.82	3.19	12.64	3.49	12.84	3.51	12.68	3.61	0.01	1.00	0.00	
FFM (kg)	40.98	4.09	44.87	3.94	45.39	3.02	49.53	4.63	12.02	0.00	0.41	
TBW (kg)	29.84	3.50	31.63	3.01	32.22	3.17	34.97	3.50	7.00	0.00	0.29	
ECW (kg)	13.55	1.29	13.96	1.15	14.19	1.23	15.16	1.44	4.77	0.01	0.22	
ICW (kg)	16.29	2.26	17.67	1.87	18.04	1.96	19.81	2.10	8.42	0.00	0.33	
TUA (cm2)	49.23	10.28	55.73	12.57	53.23	5.63	58.12	7.47	2.91	0.04	0.14	
UMA (cm2)	34.65	7.53	40.22	8.64	38.07	4.47	43.15	6.48	4.60	0.01	0.21	
UFA (cm2)	14.58	3.57	15.51	4.67	15.16	3.98	14.97	3.78	0.10	0.96	0.01	
UFI	29.58	4.06	27.61	3.58	28.28	6.47	25.75	5.49	1.78	0.16	0.09	
TCA (cm2)	92.70	11.86	97.66	14.94	105.72	16.43	102.73	10.81	2.42	0.08	0.12	
CMA (cm2)	75.05	9.63	80.65	13.27	86.53	13.82	87.06	9.33	3.72	0.02	0.18	
CFA (cm2)	17.65	4.59	17.01	4.61	19.19	6.04	15.66	5.00	1.27	0.29	0.07	
CFI	18.92	3.96	17.46	3.94	18.04	4.79	15.16	4.38	2.53	0.07	0.13	
Total SK Santos (mm)	75.64	14.97	75.43	11.86	78.97	11.39	72.86	16.29	0.43	0.73	0.02	
Appendicular SK Santos (mm)	46.14	9.32	46.13	8.59	45.72	8.15	44.12	10.27	0.19	0.91	0.01	
Lower limb SK Santos (mm)	26.18	5.47	26.37	4.45	26.77	5.24	25.86	5.81	0.07	0.98	0.00	
Trunk SK Santos (mm)	29.50	6.55	29.30	4.55	33.25	6.62	28.74	6.97	1.19	0.32	0.06	
CMJ (cm)	24.51	2.37	29.97	5.03	27.88	4.19	32.00	3.92	10.67	0.00	0.41	
30–15 IFT (Km/h)	16.86	1.00	17.43	1.02	17.30	0.67	18.71	0.50	10.53	0.00	0.50	
Note:

SD, standard deviation; exp, experience; F, Snedecor-Fisher test; p, probability value; η2, partial coefficient of determination; BM, body mass; trunk h, trunk height; arm rel, arm relaxed; arm stret, arm stretched; SK, skinfold; R, resistance; Xc, capacitive reactance; PhA, angle phase; D, body density; BF, body fat percentage; FM, fat mass; FFM, free fat mass; TBW, total body water; ECW, extracellular water; ICW, intracellular water; TUA, total upper area; UMA, upper muscle area; UFA, upper fat area; UFI, upper fat index; TCA, total calf area; CMA, calf muscle area; CFA, calf fat area; CFI, calf fat index; CMJ, countermovement jump; IFT, intermittent fitness test.

Compared with the first team, the athletes U15 showed significantly lower age at menarche, stature, weight, lengths, diameters, and circumferences, while the U17 and U19 showed comparable values. Only for trunk length and hip circumference, the first teams differed from all the other groups. Skinfold thickness did not differ among groups.

The main differences were connected to body composition: FFM and body density were higher in the first teams than in U15 and U17. The data are within the range proposed for female soccer players, according to which body fat percentage ranges between 18.0% and 28.2% in players aged 10 to 16 years and from 14.5% to 22% for elite adult players (Malina et al., 2021; Randell et al., 2021).

U15 female athletes also showed lower water compartments (TWB, ECW and ICW) values, such as of TUA, UMA, CMA than first team.

The sum of the seven skinfolds (triceps, subscapular, biceps, supra iliac, abdominal, thigh, and medial calf) and sums of appendicular and trunk skinfolds are at the 25th percentile of the reference for female soccer players proposed by Santos et al. (2014). Apart from U15, arm circumference is at 50th percentile of the references. Calf and thigh circumferences are at the 50th percentile in all the groups, apart from U19 and the first team, which presented a thigh circumference at the 75th percentile.

Concerning BIA, resistance decreased with the increase of the age category, while PhA increased: significant differences were observed for PhA between the first team and the others, while for R between the two oldest and the two younger groups.

Figure 1 shows the BIVA confidence ellipses of female soccer athletes, which are calculated according to the different groups. The slope of the vector increases as the category increases; furthermore, the first team shows more homogeneous values than the other groups.

Figure 1 BIVA profiles for elite Italian female soccer players from U15 to first team.

Performance parameters (CMJ and MRS30-15IFT) were better in the first team, which significantly differs from all the teams in speed ability. The countermovement jump is significantly lower in U15 compared to U17 and the first team.

CMJ showed strong linear correlation with the sum of all skinfold thicknesses (r = −0.479, p < 0.001), majorly affected by medial calf skinfold (r = −0.475, p < 0.001) and abdominal skinfold (r = −0.466, p < 0.001), while MRS30-15IFT better related with Santos skinfold sum (r = −0.44, p = 0.008), greatly affected by supraspinal (r = −0.5, p = 0.002) and triceps skinfolds (r = −0.438, p = 0.009). The BIA phase angle reported high correlation with both CMJ (r = 0.605, p < 0.001) and MRS30-15IFT (r = 0.55, p < 0.001), even if fluid content was significantly only between ICW and CMJ (r = 0.287, p = 0.041). However, the body density was the body composition parameter that better related with performance (CMJ r = 0.548, MRS30-15IFT r = 0.435), while fatness indices of calf and arm strongly related with CMJ (CFI r = −0.522, UFI r = −0.506) and MRS30-15IFT (UFI r = −0.465, CFI r = −0.409).

The model derived from the stepwise backward multiple regression analysis (Tables 2 and 3) explained 55% of the variance in the CMJ and MRS30-15IFT tests, respectively. Training hours emerged as the most significant positive predictor for both tests, as indicated by the positive β coefficient. The unstandardized coefficient (B) (not shown in the table) for training hours in the multivariate model was 0.73, suggesting that an additional 0.7 h of training is associated with a 1 cm improvement in CMJ performance. Likewise, an increase of approximately 0.29 h of training was associated with a 1 km/h improvement in MRS30-15IFT performance. Additionally, the sum of skinfolds was a significant negative predictor for both tests. The unstandardized coefficient (B) (not shown in the table) for skinfold thickness was −0.13, indicating that a reduction of 0.13 mm in total skinfold thickness would result in a 1 cm increase in CMJ height. Similarly, a reduction of 0.03 mm was associated with a 1 km/h improvement in MRS30-15IFT performance. Furthermore, an increase of 1.74 in PhA was linked to a 1 cm improvement in CMJ performance.

Table 2 Backward multiple regression model that analyses the contributions of anthropometric, body composition, and impedance parameters to CMJ (dependent variable).

Variables	β	CI (95%)	t	p	Tolerance	R2	Adj.-R2	p	
Training hours	0.38	[0.12–0.63]	2.99	0.004	0.55	0.58	0.55	<0.001	
PhA	0.28	[0.02–0.53]	2.13	0.038	0.54				
Total skinfolds	−0.39	[−0.58 to −0.19]	−3.99	0.000	0.96				
Note:

β, standardized regression coefficient; CI, confident interval; t, t-statistic for the significance of the regression coefficient; p, probability value; R2, coefficient of determination; Adj. R2, adjusted R2; PhA, angle phase.

Table 3 Backward multiple regression model that analyses the contributions of anthropometric, body composition, and impedance parameters to MRS30-15IFT test (dependent variable).

Variables	β	CI (95%)	t	p	Tolerance	R2	Adj.-R2	p	
Training hours	0.64	[0.40–0.88]	5.44	0.000	0.95	0.58	0.55	<0.001	
Total skinfolds	−0.30	[−0.53 to −0.06]	−2.52	0.017	0.95				
Note:

β, standardized regression coefficient; CI, confident interval; t, t-statistic for the significance of the regression coefficient; p, probability value; R2, coefficient of determination; Adj. R2, adjusted R2.

Discussion

The primary aim of this study was to describe and compare the anthropometric characteristics and physical performance of female soccer players across different age groups: under 15 (U15), under 17 (U17), under 19 (U19), and the first team. We considered players’ age at menarche self-reported, training amount and soccer experience in order to justify expected differences.

The age at menarche of the athletes of the present study ranged from 11.7 to 13.6 and was lower in the younger group than in the older. The values of the present sample are reported for several samples of late adolescent and young adult soccer players (mean ages menarche ranged from 12.7–13.0 years) by Malina et al. (2021).

Though limited, the distribution suggested a trend towards somewhat earlier sexual maturation in the younger players, which is generally consistent with non-sportive girls reported by Gualdi-Russo et al. (2022) for girls in the same region (Emilia-Romagna) (11.7 years). This data was slightly lower than the mean value reported in a previous study carried out in the same region (11.97 ± 0.94 years) (Toselli et al., 2021) and for girls born approximately sixty years earlier (11.9 ± 1.4 years) always in the same region (Gualdi-Russo et al., 2022).

It should be noted that age at menarche was self-reported and that in the present study, age at menarche increases with the age of the athletes; studies that rely on self-reported data from adult women are less reliable (Cooper et al., 2006).

The age at which the athletes began soccer practice does not differ between the different groups. However, obviously, it influences the years of accumulated experience, and the hours of training expected, which differ significantly.

As regards anthropometric parameters, stature and weight of the female youth soccer players of the present study are within the range indicated for elite players aged 15–17 by Randell et al. (2021). The anthropometric characteristics of the U15 athletes significantly differed from those of the first team. At the same time, the U17 and U19 already showed comparable values, demonstrating how, with the maturation process, the desired characteristics are achieved. These data are by Ramos et al. (2021), who affirmed that the anthropometric parameters stabilize as players approach adulthood.

Body fat percentage did not differ among groups. A certain amount of fat is required to maintain body metabolism, but excess adiposity negatively influences performance, especially in soccer (Adhikari & Nugent, 2015). Excessive body fat has been linked to reduced athletic performance and increased injury risk, as it may impair movement efficiency and place additional strain on the musculoskeletal system (Oliveira et al., 2021).

In the present study, the percentage fat varied from 23.5 of U15 to 20.12 of the first team and is within the range proposed for female soccer players, according to which body fat percentage ranges between 18.0% and 28.2% in players aged 10 to 16 years and from 14.5% to 22% for elite adult players (Randell et al., 2021). Wilmore, Miller & Pollock (1974) reported an average of 22% body fat for senior soccer players, while the female soccer players studied by Adhikari & Nugent (2015) presented an average value of 22.1 ± 3.1% body fat. U15 presented lower total and muscle limb area values (TUA, UMA, CMA) than the first team.

Fat-free mass and body density were higher in the first teams than in U15 and U17. FFM ranged from 40,98 to 49,53, and Ramos et al. (2021) highlighted that female soccer players aged 19–26 years have a lean mass ranging from 42.5 to 49.5 kg, a notably higher value than that observed in younger age groups, indicating an increase in lean mass with age and training. Collegiate and professional female soccer players often experience gradual fat-free mass (FFM) increases over their careers, reflecting adaptations to advanced training stimuli (Katona et al., 2020; Minett et al., 2017). Increased fat-free mass (FFM) enhanced training intensity as players progress in competitive levels (Katona et al., 2020).

The amount of FFM bond to fluid body contents and water distribution. In fact, if no change in fat and fat-free components appears, it is expected that total body water decreases with ageing (Kyle et al., 2015). Whilst no differences emerged in female players body fat, the higher level of fat-free tissue detected in older categories may reflect linear variation in TBW and its cellular distribution. According to this physiological rationale, the U15 female soccer players showed lower values of water compartments (TWB, ECW, and ICW), which also emerged from PhA and BIVA confidence ellipses. PhA and BIVA can be used together to indicate cell integrity and hydration. The vector length in the BIVA confidence graph is influenced by the amount of total body water and fat-free mass. PhA is influenced by soft tissue (which changes depending on age and clinical conditions), hyperhydration, and fat mass (which leads to a progressive shortening of the vector) (Piccoli, Pillon & Dumler, 2002). Proper hydration, reflected in total body water (TBW) and intracellular water (ICW), is crucial for maintaining muscle function and reducing injury risk during intense training sessions (Oliveira et al., 2021). In the present study, different ellipses were observed with age: the slope of the vector increases as the category increases; furthermore, the first team presents a more compact and homogeneous ellipse compared to the other groups. This result emphasizes the difference in body composition with age and is particularly important, as they are the first to consider differences in bioelectrical parameters with age in female soccer players and can be used as a reference. Understanding body composition is vital for tailoring training programs that enhance performance while promoting lifelong health in female soccer athletes. Body composition plays a critical role in female soccer players’ performance, health, and development. Understanding its variations during youth and adulthood allows for tailored training and nutrition strategies that optimize athletic performance while safeguarding long-term health. From youth to adulthood, it is a key metric for evaluating physical readiness and mitigating risks associated with high-intensity sports participation (Lesinski et al., 2017; Oliveira et al., 2021).

The first team’s performance parameters (CMJ and MRS30-15IFT) were significantly better. The studies on CMJ performance in female soccer players highlight the influence of training interventions, strength levels, and sex-related factors (Haugen, Tønnessen & Seiler, 2012; Thomas, Jones & Dos’santos, 2022). While some studies suggest that CMJ performance improves with age and training, others emphasize the role of strength and jump strategy in determining performance outcomes.

Age-related differences in sprint performance among adolescent female soccer players were also found by Mainer-Pardos et al. (2021), who reported that older female soccer players (U16 and U18) performed better in all split sprint times than younger female soccer players (U14). Although maximal speed and change of direction ability are the most applied tests in soccer, the MRS30-15IFT may enhance the sport demands that imply several motor abilities such as endurance, power and velocity. Accordingly, the end running speed achieved with 30-15IFT reported higher linear correlation with maximal oxygen consumption (r = 0.93), maximal speed (r = 0.63) and maximal aerobic speed (r = 0.84) and power (r = 0.65; Buchheit, 2008; Dupont et al., 2010; Paravlic et al., 2022). In team sports such as soccer, the 30-15IFT is worldwide used for its complete metabolic requirement that covers both aerobic, anaerobic and neuromuscular efforts (Buchheit, 2021). We found results in accordance with previous literature, where MRS varied less than 1.3 km/h, with higher homogeneity in elite adult female players (~0.5 km/h). However, all our categories covered the specific confidence intervals detected such as 16–21.5 km/h in U16, 16–18.5 km/h in U20 and 18.5–21 km/h in senior teams (Buchheit, 2021).

The second and third purposes of our studies were to find any linear correlation between CMJ and 30-15IFT and anthropometric parameters that cover the body composition sphere, and then to predict physical performance by best regressors. As expected by sports physiology, the body fat detected by both ADP and skinfold thicknesses negatively related to jumping and running abilities, whereas body density and BIA phase angle reported positive correlations (Collins et al., 2021; Gonzalez et al., 2016; Lesinski et al., 2017). This was confirmed through backward multiple regression analysis, where training hours added a significant contribution as a predictor for both the CMJ and 30-15IFT. While it is supposed that as the training volume increases the potential physical performance gains, it still be unclear how the perfect training dosage is and what is the best body for achieving elite level. However, the design of the study here presented cannot provide any cause-and-effect relationship and more sophisticated investigations are required.

This study presents some limitations that need to be considered:

(a) Menarche age is self-reported and could be biased. However, this is a common non-invasive method in sports sciences.

(b) While we provided the total hours and days of training per week (volume), we could not account for training intensities.

(c) Participants came from the same club affecting the generalizability enhancing internal consistency but limiting the generalizability of the findings to other populations or lower-level teams. Many applications are needed.

(d) We could not analyze and compare the different field roles.

Conclusion

The findings of the present study highlight the importance of tailoring training and nutrition programs to the developmental stage of female soccer players for optimal performance and health outcomes. Coaches should employ age-specific strategies focusing on lean mass development during adolescence and maintaining optimal ratios in adulthood.

Supplemental Information

Supplemental Information 1 Dataset.

Additional Information and Declarations

Competing Interests

The authors declare that they have no competing interests.

Author Contributions

Stefania Toselli conceived and designed the experiments, performed the experiments, analyzed the data, authored or reviewed drafts of the article, and approved the final draft.

Federica Moro performed the experiments, prepared figures and/or tables, authored or reviewed drafts of the article, and approved the final draft.

Martina Perugini performed the experiments, authored or reviewed drafts of the article, and approved the final draft.

Mario Mauro performed the experiments, prepared figures and/or tables, authored or reviewed drafts of the article, and approved the final draft.

Ethics

The following information was supplied relating to ethical approvals (i.e., approving body and any reference numbers):

The study was approved by the Bioethics Committee of the University of Bologna (Approval code: 25027, March 2017).

Data Availability

The following information was supplied regarding data availability:

The raw data is available in the Supplemental Files.

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
