# Peer review of "Age-related variation in the anthropometric profiles, body composition and functional capacities of female soccer players"

_PeerJ, doi:10.7717/peerj.20096_

## Round 0.1 · original submission · Minor Revisions

· Academic Editor

Minor Revisions

Dear Authors,

Make corrections according to the reviewers' comments or write a detailed rebuttal on a point-by-point basis.

·

Basic reporting

The manuscript is clearly written in professional academic English and adheres to PeerJ’s formatting standards. The introduction provides sufficient context and presents a compelling rationale for the study. The literature is well selected and up-to-date, with strong justification for the study’s aims. Figures and tables are high in quality, relevant, and well labeled. All required raw data and supplementary material are supplied and appropriately described.
Minor Suggestion: Some terms in the Discussion (e.g., "elite body") could be reformulated for clarity and consistency with scientific writing conventions. Consider a brief language check focused on fluency and tone. The term 'motor skills' should be replaced with 'physical performance tests' in the summary and throughout the text, where appropriate. For example: line 33 physical fitness should to change with physical performance tests. The objective of the paper must be consistently stated in the abstract, at the end of the introduction, and at the beginning of the discussion. In the introduction, you mention multiple objectives, yet only one is listed. This inconsistency should be corrected. Line 99, 105, 134, 118, 146, 192, 234, 236-237, 248-249, 252, 262-264, 267, 304 – it is necessary to properly cite the reference.

Experimental design

The study is methodologically sound and meets high technical and ethical standards. It includes an appropriate cross-sectional design, examining U15, U17, U19, and first-team female soccer players within the same elite club. The sample is relatively small but well stratified by category.
The methods are described in sufficient detail for reproducibility. The use of bioelectrical impedance analysis (BIA), air displacement plethysmography (ADP), and physical performance testing (CMJ, 30-15IFT) provides a robust multidimensional approach.
Suggestions for improvement:
• Training volume is a key predictor in regression models, yet the description of training content or periodization is lacking. Please consider providing a brief summary of weekly training structure across age groups.
• Menarche age is self-reported. While this is common in field research, the limitation should be acknowledged more explicitly in the limitations section.
• Provide a reference that confirms the reliability and validity of the recommendation to eat breakfast four hours prior to bioelectrical impedance measurement.

Validity of the findings

Statistical analyses are appropriate and well presented. The use of backward stepwise multiple regression adds depth to the interpretation. The study finds that fat-free mass (FFM), phase angle (PhA), and training hours positively predict performance, while total skinfold thickness negatively correlates with CMJ and speed.
The authors interpret findings cautiously and in line with data. The findings are credible and fill an important gap in literature related to adolescent female athletes.
Minor Comment: As the study is cross-sectional, statements implying causality (e.g., "training hours improve CMJ by 1 cm") should be slightly tempered.

Additional comments

Relevant topic with practical application for talent identification and athletic development in female soccer.
Multidimensional evaluation including body composition, hydration, and performance metrics.
Use of BIVA and PhA to contextualize physiological maturity is innovative and valuable.
The sample is from a single elite club. Although this improves internal consistency, generalizability is limited. This should be addressed in the Discussion.
Positional data (e.g., defender, midfielder) was not analyzed. If available, including it could enrich the interpretation.
This is a high-quality manuscript that provides novel and practical insights into age-related physiological development in female soccer players. I recommend minor revision before acceptance. Addressing the small number of suggested clarifications would further enhance the clarity and impact of the paper.

Reviewer 2 ·

Basic reporting

Consistency in terminology: There are variations (e.g. physical perfomance > line 3, 15 and 17, often used in this article, but also terms "physical characteristics" line 7, 22, & 32 and "motor performance" line 70 nad 235. It should be defined what these terms are or use consistent terminology if they have sam meaning.


In the abstract section, it would be desirable to define what type of regression analysis is being used in the research. Line 10.

Female soccer players must possess a combination of physical and psychological characteristics to be competitive at the highest level. These characteristics are essential not only for selection processes but must also be considered for injury prevention and continuous performance improvement. Line 31-33 Instead of the term "must possess" the term "required to have" could be used

dash plumb line symbol "health-" line 39.

(Markovic & Mikulic, 2010), different text style, line 61
(Adhikari & Nugent, 2015) also line 263 & 271

Are they (fe.g. power and velocity) physiological or motor characteristics? These are motor characteristics/abilities.
Sentence"that imply several physiological characteristics such as endurance, power and velocity. Line 316 & 317"


Line 419 Referencing style> NORGAN, N. G. (1988).

Experimental design

no comment

Validity of the findings

no comment

Additional comments

It is well-organized, uses relevant literature, and provides a rich data-driven approach. The methodology is grounded, and you clearly justify the statistical method for chosen data.

---

## Round 0.2 · accepted · Accept

· Academic Editor

Accept

Dear Authors,

Your manuscript is now acceptable for publication in its current form.

·

Basic reporting

The term “motor skills” has been replaced with “physical performance tests” in the abstract, introduction, and discussion.

The expression “elite body” has been reformulated for clarity and consistency with academic style.

The research objectives are now clearly and consistently stated in the abstract, at the end of the introduction, and at the beginning of the discussion.

The missing references at lines 99, 105, 118, 134, 146, 192, 234, 236–237, 248–249, 252, 262–264, 267, and 304 have now been added and properly cited.

Experimental design

A description of the weekly training structure and volume across age groups has been added to the methodology.

In the Limitations section, it is explicitly stated that the data on menarche are self-reported, with a note on potential bias.

A reference (Kyle et al., 2004) has been added to confirm the recommendations regarding BIA measurement after breakfast.

Validity of the findings

The results and discussion have been revised to avoid causal statements. It is now emphasized that training, FFM, and PhA are predictors or associated factors rather than direct causes.

Additional comments

The Limitations section highlights the limited generalizability of the findings since the sample comes from a single club.

It is also noted that it was not possible to analyze the data by player positions.